# Microduplication 3p26.3p24.3 and 4q34.3q35.2 Microdeletion Identified in a Patient with Developmental Delay Associated with Brain Malformation

**DOI:** 10.3390/diagnostics12112887

**Published:** 2022-11-21

**Authors:** Georgeta Cardos, Nicolae Gica, Corina Gica, Anca Maria Panaitescu, Mariana Predescu, Gheorghe Peltecu, Florina Mihaela Nedelea

**Affiliations:** 1Filantropia Clinical Hospital, 011132 Bucharest, Romania; 2Department of Obstetrics and Gynecology, Carol Davila University of Medicine and Pharmacy, 020021 Bucharest, Romania

**Keywords:** SNP-array, molecular karyotyping, intellectual disability

## Abstract

Microdeletions and microduplications are involved in many of prenatal and postnatal cases of multiple congenital malformations (MCM), developmental delay/intellectual disability (DD/ID), and autism spectrum disorders (ASD). Molecular karyotyping analysis (MCA), performed by DNA microarray technology, is a valuable method used to elucidate the ethology of these clinical expressions, essentially contributing to the diagnosis of rare genetic diseases produced by DNA copy number variations (CNVs). MCA is frequently used as the first-tier cytogenetic diagnostic test for patients with MCM, DD/ID, or ASD due to its much higher resolution (≥10×) for detecting microdeletions and microduplications than classic cytogenetic analysis by G-banded karyotyping. Therefore, MCA can detect about 10% pathogenic genomic imbalances more than G-banded karyotyping alone. In addition, MCA using the Single Nucleotide Polymorphism-array (SNP-array) method also allows highlighting the regions of loss of heterozygosity and uniparental disomy, which are the basis of some genetic syndromes. We presented a case of a five-year-old patient, with global development delay, bilateral fronto-parietal lysencephaly, and pachygyria, for which MCA through SNP-Array led to the detection of the genetic changes, such as 3p26.3p24.3 microduplication and 4q34.3q35.2 microdeletion, which were the basis of the patient’s phenotype and to the precise establishment of the diagnosis.

Anomalies (deletions and duplications) of the short arm of chromosome 3 are rare and their clinical significance is still incompletely elucidated. Te Weehi et al. (2014) reported a complex rearrangement of a 913 kb within the 3p26.3 region consistent with duplication (encompassing the amino-terminal regions of the *CHL1* and *CNTN6* genes, respectively) and its correlation with neuro-development in a 34-month-old boy with global developmental delay and ASD and compared the findings with other case studies reported elsewhere. Some other genes have been suggested to be involved in the phenotypic expression of chromosomal abnormalities in the 3p26.3 region. Among these, *CRBN* and *CNTN4* genes are thought to account for dysmorphic features and intellectual disability (being suggested to cause typical 3p deletion syndrome) [1]. The *CHL1* gene, which encodes a protein that belongs to the L1 gene family of neural-adhesion molecules that regulate brain cell migration and synaptogenesis highly expressed in the central and peripheral nervous systems, seems to be involved in cognitive and language impairments in both deletions and duplications of the 3p26.3 region [1,2,3].

**Figure 1 diagnostics-12-02887-f001:**
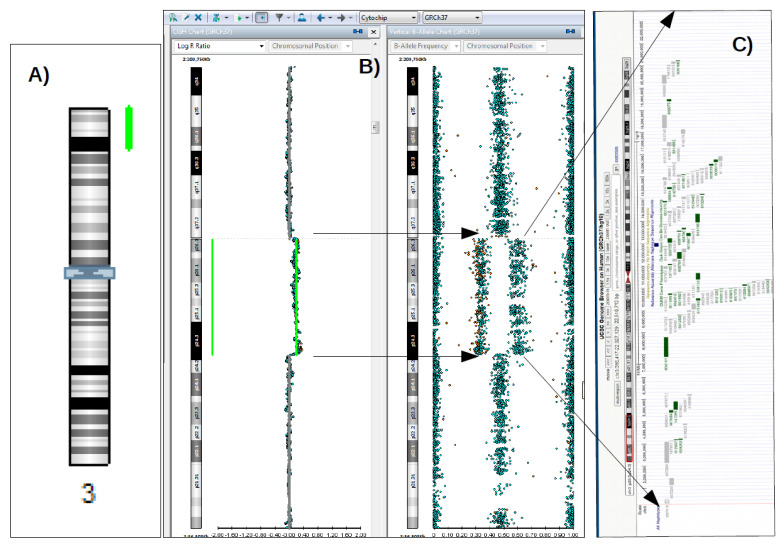
Pathogenic duplication of 22.5 Mb in 3p26.3p24.3 region detected by SNP-Array (green lines in “(**A**,**B**)” in a five-year-old patient, with delay of psycho-motor development, bilateral fronto-parietal lysencephaly, and pachygyria. SNP-Array methodology: the DNA sample isolated from peripheral blood was analyzed with the HumanCytoSNP-12 v2.1 Analysis BeadChip Kit (Illumina). The scanning was performed with the NextSeq550 equipment and the software related to the equipment (Illumina). Data analysis was performed with BlueFuse Multi 4.5 Software (32178) (Illumina), using the databases: UCSC Genome Browser, DECIPHER, OMIM, ISCA, DGV, ClinGen and ClinVar. The patient is a female child of healthy non-consanguineous parents. There is no family history of developmental delay. Molecular karyotype formula (according to ISCN 2016): arr[GRCh37]3p26.3p24.3(316417-22827129)x3,4q34.3q35.2(182338549-190880409)x1; “(**C**)”Schematic representation of genes localized in the duplicated region 3p26.3p24.3, located in the red rectangle on the schematized chromosome 3 (from UCSC browser). Each rectangle represents an OMIM gene, those colored in green are genes with known involvement in pathogenesis. The 22.5 Mb duplication detected in the 3p26.3p24.3 region contains 80 OMIM genes, many of them being candidate or associated with pathogenesis, such as the *SETD5* gene (OMIM 615743, associated with intellectual disability, autosomal dominant AD 23, OMIM 615761), *CRBN* gene (OMIM 609262, associated with intellectual disability, autosomal recessive, AR, 2, OMIM 607417), *CCDC174* gene (OMIM 616735, associated with Hypotonia, infantile syndrome, with psychomotor inhibition, autosomal recessive AR, OMIM 616816), *BRPF1* gene (OMIM 602410, associated with Intellectual developmental disorder syndrome with dysmorphic facies and ptosis, AD, OMIM 61733), *CHL1* and *CNTN6*, those being ASD candidate genes, playing an important role in language and cognitive development [1,2]. More other microduplications of comparable size, with (likely) pathogenic significance, were reported in clinical databases, such as ClinVar (Variation ID: 155700, 148876, 57977) and Decipher (ID patients: 400840, 292119) in case of patients with intelectual disability, MCM (such as: tetralogy of Fallot, abnormality of the genitourinary, digestive, and/or musculoscheletal system). Additionally, 3q26 microduplication syndrome is described in Orphanet database as a syndromic form associated with prenatal and postnatal growth inhibition, developmental delay, intellectual impairment, dysmorphic signs, and variable combination of congenital anomalies, including cardiovascular, genitourinary, and skeletal anomalies and spectrum of caudal malformations (ORPHA:96095).

**Figure 2 diagnostics-12-02887-f002:**
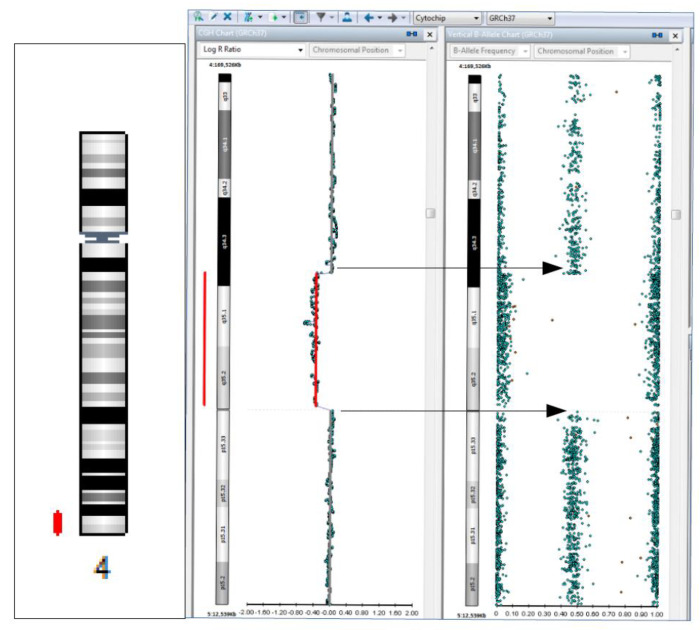
Deletion of 8.5 Mb (in chromosomal regions marked with red lines), with pathogenic significance, detected in the 4q34.3q35.2 region, containing 21 OMIM genes. The deletions comprising the 4q31q35 region have been known and reported in clinical databases and scientific literature as being responsible for phenotypic manifestations corresponding to the 4q terminal deletion syndrome (or distal monosomy 4q), including craniofacial anomalies, dysmorphic features, intellectual disabilities, developmental delay, ocular, cardiac, genitourinary malformations, and pelvic/limb dysmorphism (ORPHA:96145) [4,5]. Based on all this evidence, both 3p26.3p24.3 duplication and 4q34.3q35.2 deletion detected in the patient’s sample were classified as pathogenic CNVs [4], and contributed to the patient’s phenotypic expression; their simultaneous presence in the case of a single patient has not been reported until now, to our knowledge. The two detected genomic changes could be associated with an unbalanced translocation with a possible parental origin.

Therefore, the classic cytogenetic investigation for both the patient and her parents is necessary, as well as the confirmation by other methods (such as FISH, MLPA) of the results obtained by SNP-array. In our case, the absence of those tests represents the limits of the case study presented in this paper.

As a conclusion, because chromosomal anomalies are one of the most important causes of DD/ID and conventional karyotyping has limitations due to its low resolution, having a detection rate of only 3–5%, cases of DD/ID in patients who have normal karyotype results are still unexplained [6]. Chromosomal microarray techniques have improved the detection of small genomic deletions and duplications (CNVs) that are not routinely detected with karyotyping. Chromosomal microarray analysis can identify genomic changes responsible for ID/DD, congenital malformations, and autism, which cannot be diagnosed with classical karyotype, and can increase the diagnosis of those cases in an additional 12% to 15% of affected children. [4,6,7]. In this context, using the SNP-Array method, applied to adequate indications, increases the chance to find underlying genetic cause. Once the diagnosis is established, the optimum management of the case and eventual specific treatments and care could be assessed. More than that, accurate familial recurrence risk is available.

## Data Availability

Data analysis was performed with BlueFuse Multi 4.5 Software (32178) (Illumina), using the databases: ClinGen and ClinVar, DECIPHER, DGV, ISCA, OMIM, UCSC Genome Browser.

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
