# Peer review of "Microduplication 3p26.3p24.3 and 4q34.3q35.2 Microdeletion Identified in a Patient with Developmental Delay Associated with Brain Malformation"

_diagnostics, 2022, doi:10.3390/diagnostics12112887_

Round 1
Reviewer 1 Report
The authors describe a child with brain malformations and developmental delay found by SNP microarray to have two copy number variations that are apparently causal. I feel that this paper needs much revision if it is to be suitable for publication. I have the following recommendations:
- The idea that microarray can increase diagnostic yield above karyotype is not new. Why is this case unique? These deletions and duplications seem quite large. I believe karyotype would have detected them?
- These copy number variations have been described previously, but the authors do not discuss what their case adds to our understanding of these CNVs or provide a table summarizing these CNVs, which would add to the literature beyond another example of congenital anomalies associated with these CNVs
- It is spelled "lissencephaly"
- Molecular karyotyping I believe refers to array CGH, which is not the same thing as a SNP array and does not detect uniparental disomy or copy number neutral loss of heterozygosity
- The UCSC browser images are too small and the genes can't be seen. I recommend making a schematic
- The discussion in the figure legends should be moved to the text
Author Response
Thank you for your comments!
- The idea that microarray can increase diagnostic yield above karyotype is not new. Why is this case unique? These deletions and duplications seem quite large. I believe karyotype would have detected them?
Response: The SNP-array method for molecular karyotyping was the first approach in the investigation of the presented case, following the international recommendations, as below:
“Available evidence strongly supports the use of CMA in place of G-banded karyotyping as the first-tier cytogenetic diagnostic test for patients with DD/ID, ASD, or MCA. G-banded karyotype analysis should be reserved for patients with obvious chromosomal syndromes (e.g., Down syndrome), a family history of chromosomal rearrangement, or a history of multiple miscarriages.” (Miller et al, 2010). Concerning the deletion and duplication detected, their simultaneous presence in the case of a single patient has not been reported until now, to our knowledge.
Regarding the blood karyotypes, we proposed to family to come for testing (including the parents), but they are not from Bucharest and will come as soon as they can.
References
- Miller, D.T.; Adam, M.P.; Aradhya, S.; Biesecker, L.G.; Brothman, A.R.; Carter, N.P.; Church, D.M.; Crolla, J.A.; Eichler, E.E.; Epstein, C.J.; Faucett, W.A.; Feuk, L.; Friedman, J.M.; Hamosh, A.; Jackson, L.; Kaminsky, E.B.; Kok, K.; Krantz, I.D.; Kuhn, R.M.; Lee, C.; Ostell, J.M.; Rosenberg, C.; Scherer, S.W.; Spinner, N.B.; Stavropoulos, D.J.; Tepperberg, J.H.; Thorland, E.C.; Vermeesch, J.R.; Waggoner, D.J.; Watson, M.S.; Martin, C.L.; Ledbetter, D.H. Consensus statement: chromosomal microarray is a first-tier clinical diagnostic test for individuals with developmental disabilities or congenital anomalies. Am J Hum Genet. 2010 May 14;86(5):749-64. doi: 10.1016/j.ajhg.2010.04.006. PMID: 20466091; PMCID: PMC2869000.
- It is spelled "lissencephaly"
We are apologizing for this mistake, thank you.
- Molecular karyotyping I believe refers to array CGH, which is not the same thing as a SNP array and does not detect uniparental disomy or copy number neutral loss of heterozygosity.
Response: “Molecular karyotyping, also “array-comparative genomic hybridization” (aCGH) or “chromosomal microarray” (CMA), is a major approach used in human genetic diagnostics and cytogenomic research. Its increasing importance for human genetic diagnostics is discussed, which is mainly due to single nucleotide polymorphism (SNP) based CMAs being able to detect not only chromosomal imbalances but also isodisomy of larger DNA stretches”.
Ref: Anja Weise, Thomas Liehr, Chapter 6 - Molecular karyotyping, Editor(s): Thomas Liehr, Cytogenomics, Academic Press, 2021, Pages 73-85, ISBN 9780128235799, https://doi.org/10.1016/B978-0-12-823579-9.00006-0. (https://www.sciencedirect.com/science/article/pii/B9780128235799000060)
- The UCSC browser images are too small and the genes can't be seen. I recommend making a schematic
Response: We considered that the most relevant genes associated with pathogenesis have already been mentioned in the legend of the figure, the image is illustrative only to emphasize the density of genes in the chromosomal region involved in CNV changes with pathogenic significance for the patient.
- The discussion in the figure legends should be moved to the text
We have understood to write more details regarding images in the legend, but it is no possible to transfer it in the text.
- These copy number variations have been described previously, but the authors do not discuss what their case adds to our understanding of these CNVs or provide a table summarizing these CNVs, which would add to the literature beyond another example of congenital anomalies associated with these CNVs
Response: Because our article was submitted as Interesting Images, not as a full article, we didn”t consider necessary to mention all those aspects.
Reviewer 2 Report
Although the topic is of interest, however there are several essential improvements and clarifications that can be made in order to consider publishing of this paper.
1) The title is too general and needs to be more specific and relevant in line with the major topic of this manuscript.
2) Authors should express more clearly the relevance of molecular karyotyping in testing of patients with unexplained ID and/or multiple congenital anomalies (MCA).
3) It would be interesting if the authors would add some data about the limitations of the SNP Array method. Please offer an explanation about the limitation of the SNP arrays regarding the probe selection.
4) The methodology lacks important data! The authors should add few information on the array platform and the software used for visualizing of SNP array data (detection algorithm).
5) Did the authors analyse the samples obtained from the parents? Why not?
6) Why the authors didn’t use a confirmation method? Can the authors offer a justification in this “Interesting Images”.
7) I think the data would benefit it the authors included more appropriate references related to their results.
Author Response
Thank you for your comments!
1) The title is too general and needs to be more specific and relevant in line with the major topic of this manuscript.
Response: Thank you for suggestion, we can propose the title:
“Microduplication 3p26.3p24.3 and 4q34.3q35.2 microdeletion identified in a case with developmental delay associated with brain malformation.”
2) Authors should express more clearly the relevance of molecular karyotyping in testing of patients with unexplained ID and/or multiple congenital anomalies (MCA).
Response: Because chromosomal anomalies are one of the most important cause of DD/ID and conventional karyotyping have limitation due to low resolution, having a detection rate of only 3–5%, still are unexplained cases of DD/ID in patients who have normal karyotype results. [1].
Chromosomal microarray techniques has improved the detection of small genomic deletions and duplications (or copy-number variants) that are not routinely seen on karyotyping. Chromosomal microarray analysis can identify genomic changes responsible of ID/DD, congenital malformations, autism, that cannot be diagnosed with karyotype, and can increase the diagnosis of those cases in additional 12 to 15% of affected children. [1,2,3].
- Shevell M, Ashwal S, Donley D, Flint J, Gingold M, Hirtz D, et al. Practice parameter: evaluation of the child with global developmental delay: report of the Quality Standards Subcommittee of the American Academy of Neurology and The Practice Committee of the Child Neurology Society. 2003;60:367–380. [PubMed]
- Miller DT, Adam MP, Aradhya S, et al. Consensus statement: chromosomal microarray is a first-tier clinical diagnostic test for individuals with developmental disabilities or congenital anomalies.Am J Hum Genet 2010;86:749-764
- Manning M, Hudgins L. Array-based technology and recommendations for utilization in medical genetics practice for detection of chromosomal abnormalities.Genet Med2010;12:742-745
3) It would be interesting if the authors would add some data about the limitations of the SNP Array method. Please offer an explanation about the limitation of the SNP arrays regarding the probe selection.
Response: Because our article was submitted as Interesting Images, not as a “in extenso article”, we didn”t consider necessary to mention all those aspects.
4) The methodology lacks important data! The authors should add few information on the array platform and the software used for visualizing of SNP array data (detection algorithm).
Response: The DNA sample was isolated from peripheral blood was analyzed with the HumanCytoSNP-12 v2.1 Analysis BeadChip Kit (Illumina). The scanning was performed with the NextSeq550 equipment and the software related to the equipment (Illumina). Data analysis was performed with BlueFuse Multi 4.5 Software (32178) (Illumina), using the databases: UCSC Genome Browser, DECIPHER, OMIM, ISCA, DGV, ClinGen and ClinVar.
5) Did the authors analyse the samples obtained from the parents? Why not?
Response: We advised to complete investigations with parental karyotypes in order to establish further familial reccurence risk, but they are not from Bucharest and will come as soon as they can.
We have sometimes cases for which simple things are more complicated (sometimes are coming back in Bucharest from small villages even after one year).
6) Why the authors didn’t use a confirmation method? Can the authors offer a justification in this “Interesting Images”.
Response:
The analysis of the karyotype both in the case of the patient (as confirmation method) and the parents is the next step or stage in the analysis/elucidation of the case mentioned in the article, for a good management of the patient. The parents have been contacted (they live over 200 km away from Bucharest) and they will find a way of transport to present themselves at our laboratory for blood collection for classic cytogenetic analysis.
We also mentioned in the report the recommendations regarding those tests.
We consider it as “Interesting Images” thinking at the fact that this is a brief communication for clinicians, and for them those results are not so often seen, as the geneticist are during the analysis.
7) I think the data would benefit it the authors included more appropriate references related to their results.
Response: Thank you for the suggestion.
We may add those mentioned before (Q 2).
Round 2
Reviewer 1 Report
Thank you for your responses.
Author Response
Thank you for your comments!
Reviewer 2 Report
The authors have satisfactorily responded to my comments, with few exceptions, but considering the type of the manuscript “Interesting Imagines” I think the paper will be ready for publication only if minor improvements are made:
1. The confirmatory methods for molecular karyotyping are not limited to “the classic cytogenetic investigation” as the authors suggested. Other methods are used to confirm these kinds of results, including FISH, MLPA etc. As the authors formulated in the paper, it can be wrongly interpreted that the confirmatory methods are limited to “the classic cytogenetic investigation”.
2. The authors stated “The analysis of the karyotype both in the case of the patient (as confirmation method) and the parents is the next step or stage in the analysis/elucidation of the case mentioned in the article”. The confirmation of the results is a limitation of this paper and mandatory before publication, not a next step, but as I stated before, considering the type of the manuscript “Interesting Imagines”, I think this can be included as a limitation. Thus, please include this in the paper as a limitation of your results.
Author Response
thank you for your comments!